# Snakes Elaphe Radiata May Acquire Awareness of Their Body Limits When Trying to Hide in a Shelter

**DOI:** 10.3390/bs9070067

**Published:** 2019-06-26

**Authors:** Ivan A. Khvatov, Alexey Yu. Sokolov, Alexander N. Kharitonov

**Affiliations:** 1Center for Biopsychological Studies, Moscow Institute of Psychoanalysis, 121170 Moscow, Russia; 2Institute of Psychology, Russian Academy of Sciences, 129366 Moscow, Russia

**Keywords:** body-awareness, self-awareness, body schema, behavioral flexibility, reptiles, snakes, body limits

## Abstract

Body awareness was studied experimentally in the rat snakes *Elaphe radiata*. The experimental design required that the snakes take into account the limits of their bodies when choosing a suitable hole for penetration into the shelter. The experimental setup consisted of two compartments, a launch chamber and a shelter, separated by a partition with openings of different diameters. The diameters of the holes and/or their position in the partition were changeable. The subjects were 20 snakes divided into two groups, for one of which only the locations of the holes varied; for another, both the location of the holes and the limits of the body varied. The body was increased by feeding the snakes. In the course of the first three experimental series the snakes formed the skill of taking into account the body limits, which manifested in the reduced number of unsuccessful attempts to select holes too small for their bodies. During the fourth series, with the locations of holes randomized for each trial, the snakes demonstrated behavioral flexibility, significantly more often penetrating into the shelter from the first attempt irrespectively of the location of the suitable hole. We argue that these results demonstrate the body-awareness in snakes.

## 1. Introduction

One of the essential aspects of consciousness, no matter whether it is an animal or human, is self-awareness, i.e. the ability of the subject to separate self from the environment and oppose to it [1]. This phenomenon is rarely studied on animals and is often linked to complex forms of perspective taking and empathy [2,3].

In classic experiments to identify the prerequisites of self-awareness a mirror test was used that reveals the ability to recognize one’s own reflection in a mirror. In the experiments by G. Gallup [1], chimpanzees, under light anesthesia, put spots of paint on one of the eyebrows and on the opposite ear. After awakening, the animals did not feel any physical consequences of the manipulations performed on them: They touched these parts of the body no more than any other. Seeing themselves in the mirror, they began to actively touch the painted places. It was concluded that chimpanzees remembered their appearance and noticed changes in it, and also understood that the image in the mirror was equivalent to their own body. At present, the ability of self-recognition of pongids [1,4], dolphins [5], orcas [6], elephants [7], magpies [8] and cleaner wrasses [9] is well established. In addition, some species are able to orient themselves in the surrounding space, using a mirror as a tool (to detect hidden bait), e.g., pigeons [10] and pigs [11]. Humans demonstrate self-recognition in the mirror, starting from 1.5–2 years [12].

It should be noted that this method is subject to significant criticism. Firstly, it is difficult to apply it to animals in which visual modality is not a leading one (most mammals) [13]. Secondly, both animals and humans may not be interested in markings on their own bodies [14,15]. Thirdly, the question remains open whether self-recognition in the mirror indicates self-awareness [14].

In connection with the growing criticism of this methodological approach, more and more authors are striving to offer alternative methods for identifying the ability of self-recognition. In particular, the phenomenon of recognizing own chemical traces on surrounding objects, the so-called “olfactory mirror”, is investigated [16,17,18].

Another direction of research in self-reflection is the study of the phenomenon of the scheme of the body. When orienting in the surrounding space for the implementation of locomotion and manipulation, it is necessary to take into account the physical characteristics of own body: Limits, volume, mass—and relate them to the physical characteristics of external objects. The body schema is a model of one’s own body as a whole, including also a set of ideas about the physical characteristics of one’s body (its limits, weight, density, etc.) and its individual parts, allowing planning and carrying out various movements [19]. According to one view, the body schema (taking the self into account) is phylogenetically the earliest stage in the development of all other ideas about oneself, including self-awareness [20,21].

Mainly, modern authors are studying the features of the scheme of human body. Recent studies have shown that the body schema has high plasticity and is able to integrate external objects into its structure that are in physical contact with the individual—for example, the tools used by it [22,23,24,25,26,27,28,29]. These facts are consistent with the idea of considering a tool as a probe [21]. In accordance with the change in the body pattern, a modification of the subjective perception of the environment also occurs [23,27]. Only a few studies have been devoted to the features of the primate body schema [30,31]. The body schemata of representatives of other vertebrate species have not been studied. However, the study of the ability of various animals to take into account the physical parameters of their body seems very promising due to the above limitations of the mirror-test.

Relatively recently, researchers began to focus on the body-awareness phenomenon—the ability of animals to take into account their own body’s relationship with environmental objects and to perceive their body as an obstacle to solving various problems. There is evidence of how different animals naturally take their bodies into account or their separate parts when overcoming obstacles [32]. In the study of body-awareness in Asian elephants (*Elephas maximus*) [33] these animals were to step on a mat and take a stick attached to it with a rope, and then pass this stick to the experimenter. To do this, the elephants had to realize that their body was an obstacle to success and first remove their weight from the mat before trying to pass over the stick to the experimenter. The elephants left the mat in the test much more often than in the control group, where it was not necessary to step off the mat. A similar technique was previously used in the experiments with children [34]. Thus, it may be assumed that self-recognition in the mirror or own smell landmark recognition are particular manifestations of the body-awareness phenomenon, that is, the ability to take into account various features and functioning of own physical body during the regulation of behavior and interaction with environmental objects. Consequently, one may study the phenomenon in various ways in different species, taking into account the peculiarities of their behavior in natural conditions.

The aim of our study was the phenomenon of body-awareness in snakes, or rather, a particular aspect of this phenomenon, expressed in their ability to take into account the limits of their own body when interacting with external objects. These animals were chosen as subjects for the reason that the task of penetration into various holes, e.g., to find shelter, is natural (eco-friendly) for them. In particular, after swallowing the prey, the borders of the body of the snake increase sharply, which affects the interaction with external objects. We propose that in connection with these characteristics of vital activity, they may have the ability to perceive their body as an obstacle to solving the problem of penetration into the shelter. Accordingly, in the experiment, we simulated a situation in which a snake, previously trained to enter a shelter through a certain hole, faced the impossibility of solving this problem in the same way when the experimenter changed the conditions in one of the two ways: Either by reducing the size of the hole, or by increasing the limits of the snake’s body (after feeding).

Traditionally, reptiles are considered to have limited cognitive abilities in comparison with mammals and birds [35]. However, modern experimental data indicate that they have cognitive abilities similar to those found in warm-blooded vertebrates [36,37,38,39,40,41,42]. In particular, it was shown that reptiles are capable of transferring previously acquired experience to a new situation when solving a differentiation problem [37,38,40], and are also able to form a skill through imitation [36,39,41].

Snake learning has been well studied in its various behavioral manifestations: Avoidance of stimulation (aversive stimulus), recognition of harmful and palatable substances (noxious and palatable food recognition), spatial orientation, conditioned aversion formation, conspecific recognition, habitat recognition, and cultural transmission of stimulus recognition [35,43]. In particular, it was found that aposematic coloration enhances chemosensory learning in plains gartersnakes *Thamnophis radix* [44].

Our subjects were snakes *Elaphe radiata*. This is a rather large snake, up to 1.8 m long (adults). The body is slender, the head is small, poorly delimited from the body. The eyes are rather large. The color is light yellow, orange or bright brown. The snake inhabits various biotopes from plains to mountain forests, adheres to forest margins, often occurs along the banks of water pools, lakes, rivers, and in the fields of cultivated plants. Active during the daytime, feeds on rodents [45]. In response to a threat or when exposed to open, brightly lit spaces, the snake exhibits well-expressed escape behavior. We used juvenile individuals of this species because, unlike adults, when entering an open space, they do not demonstrate defensive behavior, but seek to hide in a shelter [45]. This aspect is essential for the organization of our experiment.

## 2. Materials and Methods

### 2.1. Animals

Twenty copperhead rat snakes *Elaphe radiata*: Ten males and ten females (age 3–5 months, snout-vent length: 48.5–60.5 cm, weight 35–75 g before feeding). All snakes were born and raised in the laboratory. Before the experiment, they were contained, one by one, in horizontal terrariums 700 × 400 × 500 mm with 12:12 h light:dark cycle. Soil: Pine bark-mulch, sphagnum, coconut crumb. Temperature in a warm corner was up to 31 °C, the background temperature 26–28 °C, overnight 20–24 °C; humidity 60%–80%. The animals were fed on rodents. All experimental snakes were naive.

### 2.2. Experimental Setup

The experimental setup comprised a glass terrarium with two compartments (Figure 1). Two identical experimental setups were used. Compartment A: Start chamber 500 × 390 × 500 mm. The floor was covered with marble pebbles. Compartment B: “wet” chamber (shelter) 500 × 390 × 500 mm. Pine bark mulch, sphagnum and coconut crumb were used as a substrate. The substrate was moistened initially and then additionally every two days. In this compartment a drinking bowl was located. The exterior walls and ceiling of the compartment were painted with black opaque paint to constantly maintain a low level of illumination.

The experimental setup was located in an isolated room (with an area of 5 sq. m.) in order to exclude the effect of unaccounted variables. The compartments were separated by a glass partition, also painted black. In the partition there were three round holes with a diameter of 70 mm, located 5 mm above the floor. The diameter of the holes could be varied with the help of inserts placed on the inside of compartment B.

In the experiment, we used holes of three diameters:Large hole (hereinafter L): D = 40 mm. The snakes could penetrate into this hole without difficulty even after swallowing food;Middle hole (M): Insert with a hole D = 10–14 mm (depending on the size of the snake). A snake penetrated through this hole without difficulty unless its body was enlarged by ingestion of food;Small hole (S) insert with hole D = 6–8 mm (depending on the size of the snake). The snakes could not penetrate into this hole.

We emphasize that the inserts or holes in the partition were not supplied with any identification marks (e.g., signaling dimensions or location). Previous studies have shown that interaction with an unfamiliar environment causes an increased stress in reptiles, which affects the process of learning [35,46]; Therefore, before the experiment, each snake was kept for seven days in an experimental setup with three holes of type M. It was set 12:12 h light:dark cycle. Warm spot was located in the back of compartment B. Temperature in a warm corner was up to 31 °C, the background temperature 26–28 °C, overnight 20–24 °C; humidity 60%–80%. The illumination during the light phase (when experimental series were carried out) was 300 Lx in compartment A and 30 Lx in compartment B.

### 2.3. Description of Experimental Trials

To begin the trials, the snake was placed in a foam box with a plastic bottom and moved to compartment A of the experimental setup through the back cover. Then the plastic bottom was removed, leaving the snake on the ground. Next, we kept a time interval of 30 s before starting the trial, allowing the snake to stay in the box. The box was then lifted upwards, allowing the snake to move freely inside the experimental installation. Staying on a hard substrate in a bright room devoid of any shelters was a negative stimulation for the animal, and it tried to leave compartment A and hide in compartment B (the “wet” chamber) penetrating through one of the three holes (Figure 1). The trial was considered finished when the snake made a successful attempt to penetrate into compartment B through the L or M opening, that is, when its body completely left the launch chamber. Unsuccessful attempts to penetrate into compartment B (see description below), when the snake’s body did not leave compartment A completely, were not accepted as the end of a trial. Thus, in each trial each snake made one successful penetration.

Only one experimenter worked with the snakes. Latex gloves used to interact with animals and with substrate were changed after each trial. Initially, we set the maximum trial time to 10 min, after which the trial was considered unsuccessful and a snake would be removed from compartment A. However, during the experiment it was found that all the snakes solved the problem within a maximum of 4.5 min.

Between the experimental trials within one series, a time interval of 10 min was maintained, during which the snake was kept undisturbed in compartment B (shelter). After each trial (while the snake was in compartment B), the partition between the compartments was wiped with soapy water and dried on the compartment A side to remove any possible odor trail. In addition, after each trial, the pebbles in compartment A were replaced with a new portion. In compartment B, the soil was mixed between trials when the snake was removed from this compartment through the back cover. To begin the next trial, the snake was placed again into compartment A using the previously described method.

Note that usually experimental trials are carried out with a longer time interval than in our case (two times a day) [39,47]. However, in this study, such a time interval was impossible to sustain, since the increase in the limits of the body of the animal was achieved through the feeding procedure (see below the section "Experimental design"). Accordingly, it was necessary to carry out an experimental series before the snake digested the feed. All experimental series were conducted sequentially with each snake from the two experimental groups separately that took about 72 h. Accordingly, the experiment with all animals took 60 days. The experimental procedures were conducted from 4 June 2018 to 8 September 2018.

### 2.4. Registered Indicators

We registered the number of attempts to penetrate through different types of holes separately for each hole in each trial and separately for each snake in each trial. The penetration attempts were divided into successful and unsuccessful:An attempt was considered successful if the snake penetrated entirely into the compartment B through one of the holes;Unsuccessful was considered an attempt to penetrate the hole of type S, during which the snake pressed its muzzle to this hole, making wriggling body movements. In the case of type M hole, a penetration attempt was considered unsuccessful, during which a snake with enlarged body penetrated inside this hole to the expansion point of its body (the bulk produced with the swollen mouse), after which it retreated to compartment A. Partial insertion of the head into the hole was not considered an attempt. No snakes stuck in the holes.

The experiment was recorded on a (Sony HDR-CX405) video camera located above and behind compartment A at an angle of 45 degrees. The recording of each experimental trial was reviewed separately by three experienced experimenters to ensure an independent assessment and count of the number of penetration attempts. A penetration attempt was counted if all the three experimenters independently recorded its presence. Initially, we intended to solve controversial cases by consensus, but there were no such cases.

### 2.5. Experimental Design

The experiment consisted of 4 series (Figure 2). Each series (except 4) lasted until the snake learned to penetrate into compartment B the shortest way. Criterion of learning: the snake for 5 consecutive trials does not make unsuccessful attempts thus making only one successful penetration into the compartment B.

The animals were divided in 2 groups, 10 individuals each:Experimental group 1, when the size of the holes in the partition between compartments A and B was varied;Experimental group 2, when both the sizes of the holes in the partition between holes A and B and the borders of the body of snakes (natural or enlarged) were varied.

Series 1. The task was to form a skill in snakes of both samples to penetrate into compartment B through hole No. 1.

Series 2. Conducted at the next bright phase following the end of series 1. In the interval between the end of series 1 and the beginning of series 2, the snakes stayed in the compartment B. The holes in the partition were closed. The inserts with the holes (according to the plan of the experiment) were installed immediately before the start of series 2. In group 1, the snake body limits remained unchanged, however, hole number 1 through which, in the previous series, they had formed the skill of penetration into compartment B, was made small (S), and hole No. 3 was made penetrable (M). Our task was to identify whether snakes, provided that the physical parameters of the body remained unchanged, but the environmental conditions changed, could find a new path to compartment B. In snakes of group 2 the limits of the body were increased by feeding the snake on a mouse of subadult age, due to which they were unable to penetrate entirely into the M-type opening, and, accordingly, could leave the launch chamber only through the L-type hole. Feeding was carried out in compartment No. 2, about 3 hours after the end of series 1. The swallowed food object was located in the stomach of a snake due to which the body section increased, making it impossible to penetrate into the holes of type M (see Figure 3). Our task was to identify whether snakes, due to increased limits of their body to such a size that it would be impossible to solve the previous task by the previously learned method (penetration into compartment B through hole No. 1), could modify their behavior to achieve the desired result, that is to find a new path to compartment B.

Series 3 was carried out 3 hours after the end of series 2 and provided that by the end of series 2, the subjects of both groups managed to form a new penetration skill into compartment B through hole 3. In both groups, the permeable holes were moved to the position No. 2.

Series 4 was conducted at the next bright phase after the end of series 3. The series consisted of 20 trials for each animal. In group 1 snakes, in each experimental trial, the holes were arranged randomly, provided that one of the holes has a diameter M and the others are S. We considered that snakes have the ability to take into account (be aware of) the natural limits of their body if throughout the entire series 4, they reliably make more successful attempts to penetrate the holes M. This would mean that the snakes are able to flexibly change their behavior, choosing a hole that is suitable in size to the borders of their bodies, irrespectively the position of the hole. In group 2 snakes, holes were also arranged randomly for each trial, provided that at least one of them was L, the others were M. Similarly, we believed that snakes have the ability to take into account (be aware of) the increased limits of their body, if during the entire series of 4 they reliably make more successful attempts to penetrate through the hole L.

### 2.6. Statistical Analysis

Statistical analysis was performed using Statistica version 12.6 (Dell Software, Round Rock, Texas, United States). To identify the significance of reducing/increasing number of unsuccessful attempts to penetrate into various types of holes from the first to the last sample in each series, the Wilcoxon T-test was used. To identify the significance of the differences between the number of failed attempts in group 1 and group 2, we used the Mann—Whitney U test. To identify the significance of differences in the number of successful and unsuccessful penetrations into various types of holes, the Pearson’s chi-squared test was used. In the 4th series, the obtained distributions of penetration attempts into various types of holes were compared for a total of 20 samples with a uniform distribution.

### 2.7. Ethical Note

Our study involved noninvasive observations of animal behavior which were approved by Research Ethics Committee of the Institute of Psychology, Russian Academy of Sciences (Decision 14206-2/15-32, may 17 2018).

## 3. Results

In snakes of both groups during the 1st, 2nd and 3rd series there was a decrease in the number of unsuccessful penetrations into various types of holes from the first trial to the last. In all three series, all snakes reached the learning criterion (see Table 1 and Figure 4 and Figure 5; for more details see also Appendix A).

The Wilcoxon signed-rank test was used to compare the number of unsuccessful penetrations into Compartment B (Table 2) in the beginning of the experiment (the first trial of an experimental series), and when the learning criterion was achieved by each snake of the series in the end of the experiment (signaled by the trial that ended with 5 successful penetrations without intermingling unsuccessful attempts). Indeed, different snakes reached the criterion of learning after a different number of attempts, but only the initial and the final trials were compared given that all the animals finally did learn to escape (Table 1).

Table 2 and Table 3 illustrate that by the end of the first and second series, subjects from both samples showed a significant decrease in the number of unsuccessful attempts to penetrate into compartment B (*p* < 0.01). In series 3, such a significant shift was not identified, since already in the first trial, the number of unsuccessful penetration attempts was low (both in group 1, *p* > 0.05; and in group 2, *p* > 0.01). It should be noted that a decrease in the number of unsuccessful penetration attempts in the 1st trial of series 3 was found in snakes of group 1 in comparison with the 1st trial of series 1 (Wilcoxon T-test = 2; *p* < 0.01). A similar pattern was found in group 2 (Wilcoxon T-test = 5; *p* < 0.01). On the other hand, in group 2, the first trial of series 2 showed a significant increase in the number of unsuccessful penetration attempts in comparison with the first trial of series 1 (Wilcoxon T-test = 2.5; *p* < 0.01). In addition, during the first trial of series 2, snakes from group 1 made significantly fewer unsuccessful attempts to penetrate into compartment B than the subjects of group 2 (Mann—Whitney U test = 1.5; *p* < 0.01).

It is interesting to note that at the end of the 2nd series, as well as in the subsequent series, the characteristic behavior was observed in the subjects of both groups, during which the snake crawled to different holes and drove its muzzle along the edges of the hole without making undulations in the direction of the latter. We believe that this behavior is indicative of the study by the animal of the size of holes based on the physical limits of its own body. Following this orienting behavior, the snake penetrated into one of the holes.

During series 4 (with randomization of the location of the holes) there were no significant shifts in the number of unsuccessful penetration attempts in the subjects of both samples (see Table 2). For the entire 4th series, snakes of group 1 made 22 unsuccessful attempts to penetrate, snakes of group 2 made 34 unsuccessful attempts. At the same time, no animal made more than one unsuccessful attempt for one trial.

For a statistical assessment of the prevalence of choices of penetrable holes in both samples, we used the Pearson’s chi-squared test. Based on the fact that for all 20 samples, 10 subjects of each sample made a total of 200 successful penetrations into compartment B, it turned out that in the 1st group this penetration was made from the first attempt in 178 cases and from the second attempt in 22 cases; in the 2nd group, successful penetration from the first attempt was made in 166 cases and from the second attempt in 34 cases. These empirical distributions were compared with theoretical distributions of random successful penetration into compartment B from the first attempt or after one or more unsuccessful attempts to penetrate into too small holes. The theoretical distribution was calculated on the basis that in each sample of 3 holes only one was penetrable for the body of the snake. Respectively, the probability of accidental penetration into compartment B from the first attempt was 33.3% (67 attempts). The results of the statistical analysis are shown in Table 3. From Table 3, it can be seen that in Series 4, both the snakes of group 1 and group 2 significantly more often chose a hole, suitable for their body to penetrate into the compartment B than would be expected in a random choice situation.

## 4. Discussion

The data obtained for group 1, series 1–3, indicate the reduction of number of unsuccessful attempts to penetrate into compartment B through type S-holes. Changing location of the holes first leads to an increase in unsuccessful attempts (at the beginning of series 2), but then (in series 3) snakes steadily prefer to penetrate into the compartment B through the M-holes already during the first trials of the series. In subjects of group 1, by the end of series 1, the number of unsuccessful attempts to penetrate into the S-holes also decreases. However, at the beginning of series 2 (after increasing the animal body size), the number of unsuccessful attempts to penetrate through M-holes (previously penetrable) increases significantly (as compared with the beginning of the previous series, and in comparison with the number of unsuccessful attempts to penetrate into the S-holes of the subjects of group 1 in series 2). This may be because the experimental task in series 2 for snakes from group 1 was easier, they had only one variable changed, namely the size of the holes. In snakes of group 2 in the 2nd series, both the sizes of the holes and the borders of their bodies changed (the latter were enlarged). Meanwhile, the snakes of group 2 in series 3 already demonstrate a steady preference for penetration into compartment B through L-holes: from the beginning to the end of the series, the number of unsuccessful attempts to penetrate through the M-holes is reduced slightly.

Accordingly, in the course of series 1–3, the snakes learned to penetrate into an outgrowth located in a certain part of the experimental setup (since the arrangement of the holes did not change during the series). However, by the end of series 3, the snakes of both samples had learned to recognize holes S and holes M, respectively, as impenetrable. This is evidenced, firstly, by the fact that already in series 3 from the first trial to the last there is no significant reduction in the number of unhurried attempts (considered the fact that already during the first trial there were few). Secondly, during series 4 (with randomization of the hole positions from trial to trial), the snakes of both groups were significantly more likely to penetrate into compartment B from the first attempt regardless of the location of the permeable hole.

Thus, the snakes of both groups exhibit behavioral flexibility. They transfer experience from one experimental series to another and begin to recognize specific hole sizes as penetrable to their bodies. For group 1 (natural limits of the body) M-holes are penetrable. For group 2 with the body borders enlarged after feeding, the L-holes are permeable. In series 4, with a random arrangement of holes in each trial, snakes from both groups significantly more readily (as compared with random choice) choose a permeable hole (M or L, respectively) from the very first attempt, which also suggests behavioral flexibility.

Behavioral flexibility is the ability to adjust behavior to changes in the environment [48], by directing attention to essential stimuli [49,50] and adjusting existing skills to a new problem [51]. Among reptiles, behavioral flexibility was found in turtles [52] and lizards [37,53]. In solving problems requiring behavioral flexibility, the reptiles may rely on visual reference points: Harris, Aranguren, and Bostocks [54] found that corn snakes (*Elaphe guttata guttata*), based on visual landmarks, could rapidly learn the position of a hidden goal in an open field task. The fact that in our study, snakes, before penetrating into the hole, drove the muzzle along the edges of this hole, suggests that the tactile sensations served as landmarks for them. In other words, they kinesthetically correlated the size of their bodies with the size of the hole. However, this finding needs additional experimental verification.

Meanwhile, the results of series 4 (when the location of the holes changed for each sample) suggest that snakes may develop awareness of their body limits and compare them with the size of the holes, which is the basis for behavioral flexibility in choosing a hole to penetrate and hide in the shelter. In our opinion, the awareness of the limits of the body was formed during the first 3 series of the experiment. Thus, in our experimental situation, awareness of own body in snakes is the result of learning in the process of a number of trial and error attempts. The snakes from group 1 formed the skill of taking into account the natural limits of their bodies, whereas the snakes from group 2 formed the skill of accounting for increased (due to feeding) limits of their own bodies.

In natural conditions, many snakes die, or are stuck, when encountering a hole they cannot penetrate, most probably exactly because the “learning of self-awareness” (or “taking into account the enlarged body”) requires a considerable number of trials, that is, a sufficient volume of prior experience. At least, this is one plausible explanation.

Unfortunately, the studies aimed at the awareness of own body limits in various animal species are not numerous. In similar studies conducted on skinks, it was found that these animals are also able to take into account both natural and artificially increased (using objects attached to their heads) limits of their body when choosing a suitable hole to reach the lure [55]. Amphibians (European green toads) are able to take into account only the natural, but not artificially increased, limits of their bodies when solving an escape task [56].

In earlier studies carried out on cockroaches *Periplaneta americana* [57], as well as on a number of other arthropods, it was found that representatives of this type of animals do not form the skill of taking into account either natural or increased limits of their body. When conducting a similar experiment, both by changing the location of the holes in the experimental box and by increasing the limits of their bodies, at the beginning of each new series after the previously formed skill, arthropods showed a steady increase in the number of unsuccessful attempts to penetrate into too small holes. This means that their problem solving skill is manifested only in memorizing the location of the penetrable hole (in each series separately), but not in learning that the holes of certain sizes are impenetrable for their bodies irrespectively of their location.

The results of a similar experiment on brown rats (Long-Evans) [58] suggest that these rodents do not need to learn to take into account the natural limits of their bodies: Even naive individuals practically did not make unsuccessful attempts to penetrate into holes of S-type. They formed the skill of taking into account the limits of the body much easier than reptiles, during only 3–4 trials in one series. Moreover, the rats were able to flexibly change their behavior depending on whether their body limits were enlarged or not, if in each trial of a separate series a foreign object was randomly either attached to their heads or not.

On the other hand, it is rather difficult to compare the data obtained in our experiment with the body-awareness of elephants [12] due to significant differences in the experimental methods used.

Still we believe that the methodical approach to research in body-awareness proposed in this article may be effective for use on various types of vertebrates. Our method differs significantly from the method of self-recognition in the mirror [1,13,14,15], but it also aims at body-awareness in animals, albeit is based on other effects. In addition, we believe that the proposed approach is free of some major shortcomings of the experiments with the mirror. In particular, it does not rely on the animal vision only. 

Reptiles are a key class in developing an understanding of the evolution of cognitive abilities among amniotes. The study of similarities and differences in their cognitive processes may provide new information on the homologies and analogies of cognitive mechanisms in amniotes in general [38]. The data of this study cast light on a new aspect of a comparative psychological analysis of the cognitive abilities of various animals.

## 5. Conclusion

The results of the experimental study suggest that the rat snakes *Elaphe radiata* are able to take into account both the natural limits of their bodies, and those enlarged by feeding, which makes them exhibit behavioral flexibility when choosing a suitable opening to enter the shelter.

## Figures and Tables

**Figure 1 behavsci-09-00067-f001:**
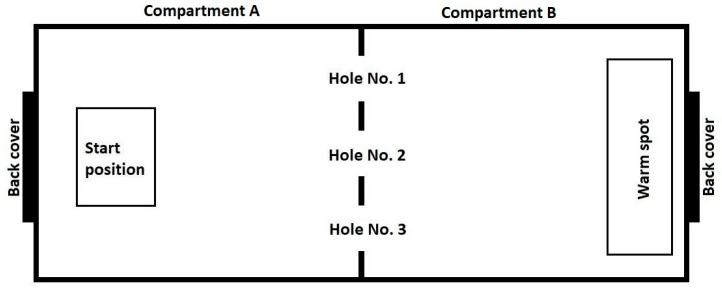
Diagram of the experimental setup (view from above).

**Figure 2 behavsci-09-00067-f002:**
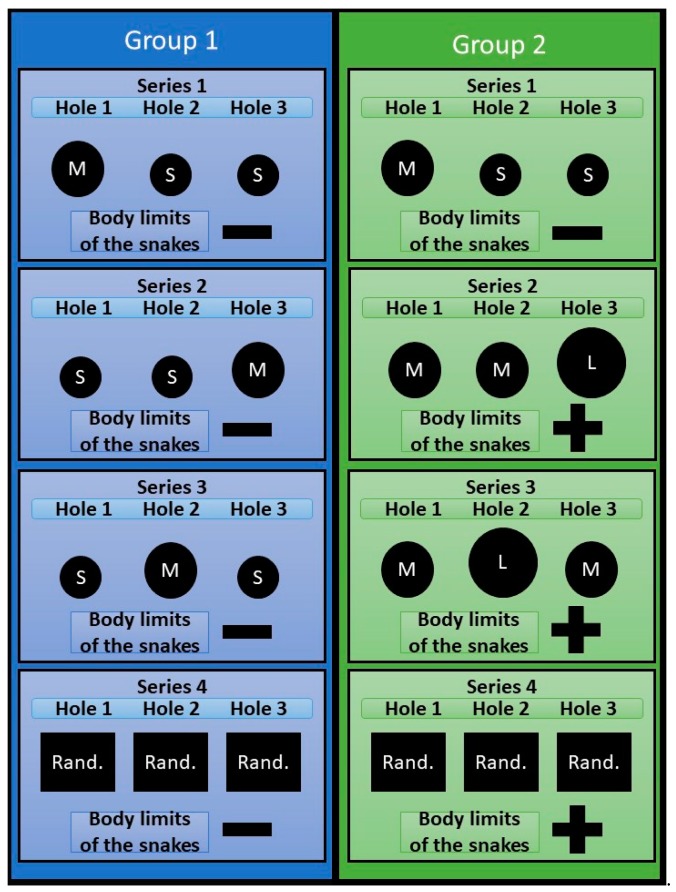
Experimental design. The dimensions and positions of the holes in each series are displayed. Body limits of the snakes: minus—natural, plus—enlarged.

**Figure 3 behavsci-09-00067-f003:**
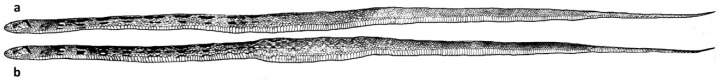
Increasing the body diameter of a snake by feeding it on a rodent: (**a**) before feeding; (**b**) after feeding.

**Figure 4 behavsci-09-00067-f004:**
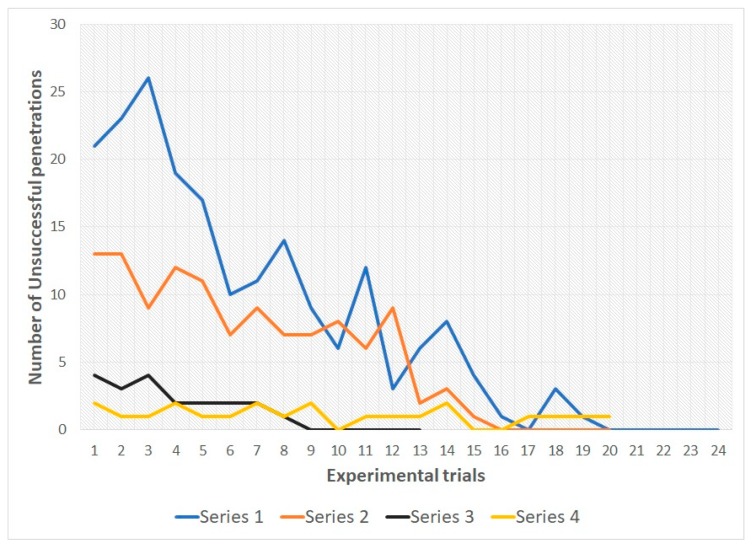
Learning curves for group 1. The total number of unsuccessful penetrations for 10 subjects is indicated.

**Figure 5 behavsci-09-00067-f005:**
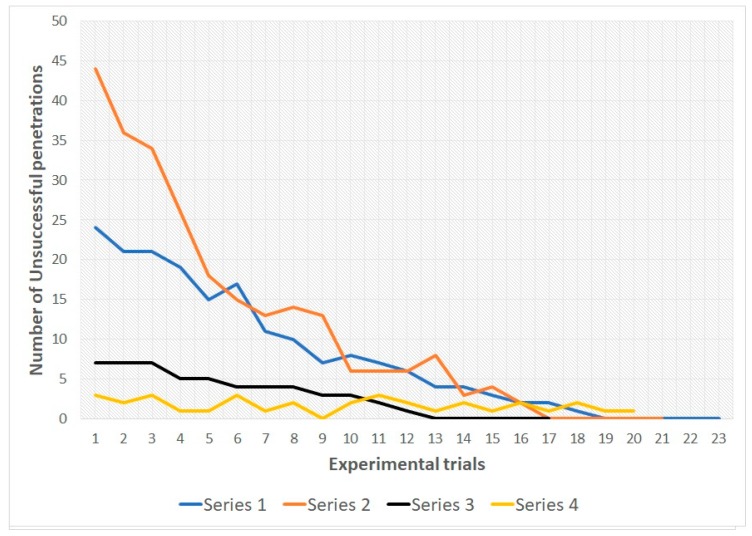
Learning curves for group 2. The total number of unsuccessful penetrations for 10 subjects is indicated.

**Table 1 behavsci-09-00067-t001:** The total number of successful and unsuccessful attempts to penetrate into compartment B in the animals of groups 1 and 2 in the first and last trials of the experimental series 1–3.

	Group 1 (10 Subjects)		Group 2 (10 Subjects)	
	Successful Penetrations	Unsuccessful Penetrations	Successful Penetrations	Unsuccessful Penetrations
Series **1**				
First trial ^1^	10	21	10	24
Last trial ^1^	10	0	10	0
The number of trials to achieve the criterion of learning ^2^	19.8 (SD = 2.25)		19.5 (SD = 2.50)	
Series **2**				
First trial ^1^	10	13	10	44
Last trial ^1^	10	0	10	0
The number of trials to achieve the criterion of learning ^2^	17 (SD = 2.21)		18.7 (SD = 2.31)	
Series **3**				
First trial ^1^	10	4	10	7
Last trial ^1^	10	0	10	0
The number of trials to achieve the criterion of learning ^2^	10.2 (SD = 1.93)		13.7 (SD = 2.21)	

^1^ Total number of penetration attempts for 10 subjects in each group. ^2^ Mean for 10 subjects.

**Table 2 behavsci-09-00067-t002:** Statistical assessment of the significance of reducing number of unsuccessful attempts to penetrate into compartment B in series 1–4 in groups 1 and 2.

	Group 1 (10 Subjects)			Group 2 (10 Subjects)		
	Total Number of Unsuccessful Penetrations	Wilcoxon T Test	*p*-Value	Total Number of Unsuccessful Penetrations	Wilcoxon T Test	*p*-Value
Series **1**						
First trial	21	1	*p* < 0.01	24	3	*p* < 0.01
Last trial	0			0		
Series **2**						
First trial	13	0	*p* < 0.01	44	0	*p* < 0.01
Lust trial	0			0		
Series **3**						
First trial	4	21	*p* > 0.05	7	10	*p* > 0.01
Last trial	0			0		
Series **4**						
Trial 1	2	37	*p* > 0.05	3	29,5	*p* > 0.05
Trial 20	1			1		

**Table 3 behavsci-09-00067-t003:** Statistical analysis of the prevalence of choices of penetrable holes in the 4th series of snakes of groups 1 and 2.

	Penetration into Compartment B from the First Successful Attempt	Penetration into Compartment B after One or More Unsuccessful Attempts	χ^2^	*p*-Value
Group 1 **(10 subjects)**				
Empirical distribution	178	22	129.780	*p* < 0.01
Theoretical distribution	67	133		
Group 2 **(10 subjects)**				
Empirical distribution	166	34	100,753	*p* < 0.01
Theoretical distribution	67	133		

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
