# Peer review of "Snakes Elaphe Radiata May Acquire Awareness of Their Body Limits When Trying to Hide in a Shelter"

_behavsci, 2019, doi:10.3390/bs9070067_

Round 1
Reviewer 1 Report
The topic is an interesting one. Many of us who work on snake behavior have experienced snakes miscalculating their body size and even making deadly mistakes. Snakes going into chicken coops and eating eggs and then not being able to escape. Snakes getting caught in bird nets protecting crops. These are two that I have experienced myself and with rat snakes, though NA species.
The experiment is an innovative way to test if snakes have an awareness of their own dimensions both before and after eating. I have some comments and suggestions.
1) I think most of the introduction is unnecessary. In fact, I would begin at line 65 and not review the mirror and olfaction work, especially all the comparative findings. If relevant, some could be brought into the discussion. If one is going to cite this work, then the studies of Chiszar, Halpin, and others on chemical self recognition should be cited.
2) Line 95 on, Although there is a selective listing of reptile learning studies, most are not relevant and I would just concentrate on snake conditioning studies, of which there are only a few, but more than listed here. Such can be located with Google Scholar and early ones in Burghardt's 1977 review of reptile learning in the Biology of the Reptilia.
3) Line 98-99. The study by Leighty et al, 2013 in the Journal of Comparative Psychology is perhaps one of the best transfer discrimination studies available in any reptile. The paper by Terrick et al., 1995 in Animal Behaviour shows multimodal conditioned aversion learning.
4) Materials and methods need more details. Length and mass of snakes pre-feeding should be provided.
5) The substrate was different in the two sides. In the start A compartment there were pebbles and spar (which is?). However, only the partition was cleaned between trials. This is not sufficient to remove odor trails. We know from an extensive series of experiments by Ludvigson that rats, also very chemically oriented, can follow trails left by other rats following rewarded versus unrewarded trials. The substrate in A should, at the least, have been thoroughly mixed up between trials and also the substrate in B should have been mixed, if not replaced (as between snakes).
4) More details on inter-trial intervals needs to be provided. Also, what was the maximum time to complete a trial? Usually some upper time limit is imposed (say 10 minutes).
5) It seems that initially the hole placement was unchanged. Thus the animals learned a spatial response only. Then, when that did not work had to search for a different hole.
6) Clarify what are the difference between series, session, and test. Needs to be in the methods prior to the results. Also, the exact dependent measures and how recorded need to be in the methods in Section 2.3. What was the operational definition of an unsuccessful penetration.
7) Who carried out the trials? One person or more than one. Were trials video-recorded? I could find no evidence that inter-observer reliability was measured or blind testing implemented. These issues need to be addressed before the ms can be accepted.
8) The results are confusing and incomplete. Do the tables represent means? Where is the learning criterion listed in the methods section? If in lines 180-84, more justification is needed as this is fairly lax.
9) The results as reported do seem convincing in terms of learning, given the limitations noted.
10) However, the fact that the animals had to learn which hole represented their body size, as compared to the rodents does suggest that in a natural setting or the ones listed at the beginning, the snakes might perish before learning a hole or entrance was too small.
11) Line 324 - first author not listed.
Author Response
We thank all the reviewers for kind attention to our work.

Reviewer 2 Report
Please see attached file.

Author Response

(The authors gave the same response as above.)

Reviewer 3 Report
The manuscript by Khvatov et al. describes the results of an interesting experiment on body size self-evaluation in snakes. I found the experiment well designed, performed, analyzed and interpreted. The authors used the similar design previously in other animals and snakes are a good group for expansion. I especially appreciate that the experiment is very natural for snakes – it is based on their natural tendency to use hollows of different size while motivated to reach a shelter.
The writing is also very good and the authors have deep knowledge of the topic.
I have some recommendations:
1) A figure somehow summarizing the rather complicated design of the experiment would be useful. The authors might try to think if a graphical summary would not be easier for readers to follow than representation of the design (and results) in the form of tables.
2) I would strongly recommend the authors to present learning curves for snakes.
3) References are not numbered according to their appearance in the text.
4) There is a typo in the table 3 („lust“) and in the figure 1 (in the word „compartement“).
Author Response

(The authors gave the same response as above.)

Round 2
Reviewer 1 Report
The manuscript has been improved by incorporating suggestions from the reviewers.
lines 54-55 - Do not see the difference between me and mine. A visual reflection is no more 'me' than a chemical reflection. I think the claim is anthropocentric
line 104 - gartersnakes, not garters
Data - All tables should list number of subjects. Also, I am confused as table 1 say they are the 'total' number of penetrations, whereas a mean would be better. But then table 2 has means and is somewhat redundant. I think only one table is needed. Also, from the supplementary file it looks as if some animals were better than others so such individual differences should be discussed. I also fear that in the Wilcoxon test used that there were ties, and thus the effective n's greatly reduced. Again, this needs explanation or reflected in the stat results. After having looked at these data again I think a statistician should evaluate whether inappropriate pooling took place.
Note that citations 35 and 44 refer to the same chapter!
prey is the word to use, not 'feed object'
The questions asked and method used are indeed interesting and novel, so would like to see the study published in some form. The fact that the prior work from the lab is not available to a non-Russian reading audience is unfortunate as the details of the prior experiments are not readily accessible.
Author Response
We are thankful to the reviewer for attention to our work.
